# On the Virasoro fusion kernel at $c = 25$

Sylvain Ribault[*] and Ioannis Tsiares[†]

Institut de Physique Théorique, Université Paris-Saclay, CNRS, CEA,
91191, Gif-sur-Yvette, France

[*] sylvain.ribault@ipht.fr , [†] ioannis.tsiares@ipht.fr

## Abstract

We find a formula for the Virasoro fusion kernel at $c = 25$, in terms of the connection coefficients of the Painlevé VI differential equation. Our formula agrees numerically with previously known integral representations of the kernel. The derivation of our formula relies on a duality $c \to 26 - c$ that is obeyed by the shift equations for the fusion and modular kernels. We conjecture that for $c < 1$ the fusion and modular kernels are not smooth functions, but distributions.

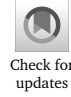

## 1   Introduction and main results

**Exactly solvable CFTs and crossing symmetry**

The properties of two-dimensional conformal field theories depend strongly on the central charge $c$ of the underlying Virasoro algebra. For low values of the central charge, there are

quite a few reasonably well-understood CFTs, such as minimal models (for rational values $c < 1$), or the critical $O(n)$ and Potts models (for all values such as $\Re c < 13$). In particular, the value $c = 1$ gives rise to a number of exactly solved theories, including compactified free bosons or orbifolds thereof [1], Runkel–Watts theory [2], and the Ashkin–Teller model [3].

For high values of the central charge $\Re c \geq 13$, the only solved, nontrivial CFTs with no extended chiral symmetry algebra are Liouville theory and generalized minimal models, which in fact exist for any $c \in \mathbb{C}$ [4]. However, the necessary conditions for high $c$ CFTs to be holographically dual to three-dimensional quantum gravity include being unitary, compact and non-diagonal [5,6]. By compact we mean that the spectrum of conformal dimensions is real, positive and discrete, with a unique state of dimension zero. By non-diagonal we mean that there are primary states with nonzero conformal spins. But Liouville theory and generalized minimal models are diagonal and non-compact.

On the other hand, there is no shortage of unitary, compact CFTs with extended chiral symmetry algebras, starting with Wess–Zumino–Witten models. The extended symmetry can then be broken by a relevant perturbation, and another CFT can be reached by following the corresponding RG flow. It is not easy to show that the resulting CFT has only Virasoro symmetry, see [7] for an argument in the case of coupled minimal models. It would be even harder to solve that CFT, where the original extended symmetry manifests itself by the presence of a large number of Virasoro representations in the spectrum.

In order to explore high $c$ two-dimensional CFTs, the most direct approach is to solve conformal bootstrap equations on various Riemann surfaces. There has been much work on the torus partition function, which so far did not result in the identification of a single CFT. One issue is that the torus partition function is not a very rich object, as it only depends on the CFT's spectrum. As a result, a modular invariant partition function does not necessarily correspond to a consistent CFT [8,9]. It would be more interesting to solve crossing symmetry for sphere four-point functions, which capture not only the CFT's spectrum but also its structure constants.

Solving crossing symmetry of four-point functions requires knowledge of the crossing properties of Virasoro conformal blocks under the fusion kernel. Even though there is no simple formula for Virasoro blocks, there is an explicit integral formula for the fusion kernel for $c \in \mathbb{C} \backslash (-\infty, 1]$ due to Ponsot and Teschner [10]. There is also another integral representation due to Teschner and Vartanov [11], which will be more useful for our purposes. For $c = 1$, a simpler i.e. non-integral expression for the fusion kernel was derived by Iorgov, Lisovyy and Tykhyy, in terms of the connection coefficient of the Painlevé VI nonlinear differential equation [12]. Using the Painlevé VI point of view, it is feasible to check crossing symmetry of four-point functions in various exactly solved CFTs at $c = 1$ [13].

**The Virasoro–Wick rotation**

Our main idea is to relate low and high values of the central charge via a map that we call the *Virasoro–Wick rotation*. Just like the fusion kernel, this map becomes simpler if we rewrite the central charge and conformal dimension in terms of variables $b$ and $P$:

$$c = 13 + 6b^2 + 6b^{-2}, \qquad \Delta = \frac{c-1}{24} - P^2. \tag{1.1}$$

We define the action of the Virasoro–Wick rotation on these parameters as

$$\begin{cases} c \to 26 - c, \\ \Delta \to 1 - \Delta, \end{cases} \iff \begin{cases} b \to ib, \\ P \to iP. \end{cases} \tag{1.2}$$

For particular objects such as fusion kernels or structure constants, the Virasoro–Wick rotation will also involve simple prefactors and/or permutations of arguments. For the moment, let

us point out that products of primary fields of the type $V_\Delta^{(c)} V_{1-\Delta}^{(26-c)}$ are commonly used in two-dimensional quantum gravity [14] and in string theory [15] in order to build generally covariant objects. What we want to do with the fields $V_\Delta^{(c)}$ and $V_{1-\Delta}^{(26-c)}$ is however not to couple them, but to compare them. In particular, let us consider the fusion kernel $\mathbf{F}_{P_s,P_t}^{(b)} \begin{bmatrix} P_2 & P_3 \\ P_1 & P_4 \end{bmatrix}$, which describes the relation between $s$- and $t$-channel four-point conformal blocks on the sphere:

$$\mathcal{F}_{P_s}^{(b),s-\text{channel}} = \int_{i\mathbb{R}} \frac{dP_t}{i} \, \mathbf{F}_{P_s,P_t}^{(b)} \mathcal{F}_{P_t}^{(b),t-\text{channel}} \,. \tag{1.3}$$

This kernel is determined by solving shift equations, which dictate its behaviour under shifts of the momentums $P_s, P_t, P_1, P_2, P_3, P_4$ by $b$ or $b^{-1}$. We find that these shift equations are invariant under a Virasoro–Wick rotation $\mathfrak{R}$, which we define as

$$\mathfrak{R} \mathbf{F}_{P_s,P_t}^{(b)} \begin{bmatrix} P_2 & P_3 \\ P_1 & P_4 \end{bmatrix} := \frac{P_t}{P_s} \mathbf{F}_{iP_t,iP_s}^{(ib)} \begin{bmatrix} iP_2 & iP_1 \\ iP_3 & iP_4 \end{bmatrix} \,. \tag{1.4}$$

We insist that it is the shift equations that are invariant, not the fusion kernel itself. The fusion kernel is an even function of the momentums, whereas its image under $\mathfrak{R}$ is odd in $P_s, P_t$. This is a priori puzzling, because the solution of the shift equations is unique, and its uniqueness does not rely on assumptions of parity [16]. As we will argue in Section 3.1, the puzzle is solved by realizing that the fusion kernel for $c \leq 1$ is not a meromorphic function of the momentums, and therefore evades the smoothness assumptions that underlie uniqueness.

Notice also that the image of a conformal block under Virasoro–Wick rotation bears no simple relation to another conformal block. This can be seen by considering the poles of the conformal block $\mathcal{F}_\Delta^{(c),s-\text{channel}}$ as a function of $\Delta$, which occur at degenerate values $\Delta = \Delta_{(r,s)}$ corresponding to momentums

$$P_{(r,s)} = \frac{1}{2}\left(br + b^{-1}s\right), \quad \text{with} \quad r,s \in \mathbb{N}^* \,. \tag{1.5}$$

The value $1 - \Delta_{(r,s)}^{(ib)} = \Delta_{(r,-s)}^{(b)}$ is not degenerate, because the indices $(r,-s)$ are no longer positive. The Virasoro–Wick rotation therefore does not map poles to poles, although it acts nicely on the residues, as can be seen by a straightforward calculation. This is somewhat analogous to the relation with harmonic analysis on a quantum group, which applies to the fusion kernel but not to conformal blocks [10,17].

**The case $c = 25$**

Finding the fusion kernel for $c < 1$ is still an open problem, and unfortunately the Virasoro–Wick rotation (1.4) of the known $c > 25$ kernel does not provide the solution. However, the situation is better for $c = 1$: from the Painlevé VI connection coefficient, we can build not only the fusion kernel, but also an unphysical object that is odd in $P_s, P_t$ [12]. Under Virasoro–Wick rotation, this object becomes even. We cannot immediately conclude that it coincides with the fusion kernel at $c = 25$, because uniqueness requires continuity in $c$, and does not apply to an object that is defined at a single value of $c$. Nevertheless, numerical comparison with the Teschner–Vartanov formula at $c = 25$ shows that there is a coincidence.

In order to write fusion kernels, we will introduce notations that make their tetrahedral symmety manifest. The tetrahedron in question, and its geometric data, are:

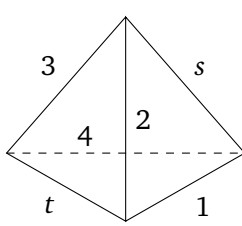

| Name | Notation | Value |
|---|---|---|
| Edges | $E$ | $\{1, 2, 3, 4, s, t\}$ |
| Pairs of opposite edges | $P$ | $\{13, 24, st\}$ |
| Faces | $F$ | $\{12s, 34s, 23t, 14t\}$ |
| Vertices | $V$ | $\{14s, 12t, 34t, 23s\}$ |

(1.6)

Formulas will involve assigning signs to edges. We use the notations:

- $\sigma \in \mathbb{Z}_2^E$ is an assignment of a sign $\sigma_i \in \{+, -\}$ for any $i \in E$, and $\sigma \in \mathbb{Z}_2^f$ for a triple of signs on a face $f \in F$.

- $\sigma_E, \sigma_v, \sigma_f$ for products of 6, 3, or 3 signs on all edges, a vertex, or a face.

- $\sigma \in \mathbb{Z}_2^E | \sigma_V = 1$ for sign assignments whose products are 1 at each vertex. There are 8 such assignments, and they can be split in two halves according to $\sigma_E = \pm 1$.

- The indicator function $\eta_i \in \mathbb{Z}_2^F$ is $\eta_i(f) = 1$ if the edge $i$ belongs to the face $f$, and $\eta_i(f) = -1$ otherwise.

Using these notations, our simpler expression for the $c = 25$ fusion kernel is

$$\mathbf{F}_{P_s, P_t}^{(c=25)} \begin{bmatrix} P_2 & P_3 \\ P_1 & P_4 \end{bmatrix} = \frac{\pi^2}{2i} \frac{G(\pm 2P_t)}{G(2 \pm 2P_s)} \prod_{f \in F} \prod_{\substack{\sigma \in \mathbb{Z}_2^f \\ \sigma_f = \eta_t(f)}} G\left(1 + \sum_{j \in f} \sigma_j P_j\right)^{-\sigma_f}$$

$$\times \sum_{\epsilon = \pm} \frac{\epsilon}{\sqrt{d}} \prod_{\substack{\sigma \in \mathbb{Z}_2^E \\ \sigma_V = 1}} \widetilde{G}\left(\omega_\epsilon - \tfrac{1}{2} \sum_{j \in E} \sigma_j P_j\right)^{-\sigma_E} , \qquad (1.7)$$

where $g(\pm x) = g(x)g(-x)$, and $\widetilde{G}(x) = \frac{G(1+x)}{G(1-x)}$, with $G(x)$ the Barnes $G$-function. The functions $d, \omega_\pm$ are defined by

$$d = \det \begin{pmatrix} 1 & -\cos(2\pi P_2) & -\cos(2\pi P_3) & \cos(2\pi P_s) \\ -\cos(2\pi P_2) & 1 & \cos(2\pi P_t) & -\cos(2\pi P_1) \\ -\cos(2\pi P_3) & \cos(2\pi P_t) & 1 & -\cos(2\pi P_4) \\ \cos(2\pi P_s) & -\cos(2\pi P_1) & -\cos(2\pi P_4) & 1 \end{pmatrix} , \qquad (1.8)$$

$$e^{2\pi i \omega_\pm} = -\frac{\sum_{ij \in P} \sin(2\pi P_i) \sin(2\pi P_j) \pm \sqrt{d}}{\frac{1}{2} \sum_{\substack{\sigma \in \mathbb{Z}_2^E \\ \sigma_V = 1}} \sigma_E e^{\pi i \sum_{j \in E} \sigma_j P_j}} . \qquad (1.9)$$

In Eq. (1.7), the appearance of the indicator function $\eta_t = -\eta_s$ breaks tetrahedral symmetry, by distinguishing the internal momentums $P_s, P_t$ from the external momentums $P_1, P_2, P_3, P_4$. That symmetry could be restored by renormalizing conformal blocks [11]. Our formulas are valid in the natural normalization

$$\mathcal{F}_{\Delta_s}^{(b), s-\text{channel}}(z) = z^{\Delta_s - \Delta_1 - \Delta_2}(1 + O(z)) . \qquad (1.10)$$

The agreement of our expression (1.7) with the Teschner–Vartanov formula boils down to the integral identity (4.7) for the function $\widetilde{G}$, which we could test numerically but not prove analytically.

Our most important numerical tests, done in Python, are found in an ancillary Jupyter notebook, which accompanies the present text on arXiv.

**Outlook**

Our results naturally lead to a few open questions:

1. *In the same way as crossing symmetry in known $c = 1$ CFTs was rederived using the relation to isomonodromic tau functions [13], can we now find exact solutions of crossing symmetry at $c = 25$?* To answer this question, it would be helpful to reformulate the analysis of [13] in terms of connection coefficients, rather than isomonodromic tau functions. We plan to work on this question in the near future.

2. *What are the modular and fusion kernels for $c < 1$, and are they distributions if $b^2 \notin \mathbb{Q}$?* In 3.1 we suggest an approach to these questions using limits from degenerate cases. Another approach would be to look for infinite series representations of the kernels as was done in [18], and to interpret divergent series as distributions. Let us also point out that for $c \geq 25$, the kernels can be viewed as scalar products of blocks from different channels [19, 20]. If a comparable scalar product could be defined for $c \leq 1$, it could help understand the analytic properties of the kernels.

3. *Is there a CFT interpretation of the van Dantzig problem?* In probability theory, the van Dantzig problem consists in finding characteristic functions $\mathfrak{f}(\vec{t})$ of a set of random variables $\vec{t}$, such that $\frac{1}{\mathfrak{f}(i\vec{t})}$ is also a characteristic function i.e. continuous, bounded and positive definite [21, 22]. Formally, the transformation $\mathfrak{f}(\vec{t}) \to \frac{1}{\mathfrak{f}(i\vec{t})}$ is similar to the Virasoro–Wick rotation, whether for the Virasoro fusion kernel (1.4) or for Liouville structure constants (3.7). Suggestively, a probabilistic construction of Liouville theory with $c > 25$ has recently emerged [23].

## 2 Shift equations and the Virasoro–Wick rotation

In this section, we consider not only the fusion kernel, but also the modular kernel. The modular kernel is simpler than the fusion kernel since it depends on only 3 momentums. We will find that the Virasoro–Wick rotation is relevant to the modular kernel as well.

The fusion kernel obeys shift equations that constrain its behaviour under shifts of the type $P \to P + b^{\pm 1}$, where $P \in \{P_s, P_t, P_1, P_2, P_3, P_4\}$. These shift equations follow from the consistency of changes of bases of five-point sphere conformal blocks, when one of the five fields is degenerate at level two [10]. Similarly, torus two-point blocks that involve one degenerate field lead to shift equations for the modular kernel [18].

We will review the shift equations for the kernels, and study their behaviour under the Virasoro–Wick rotation. In the case of the modular kernel, we will explicitly write the shift equations. In the case of the fusion kernel, we will adopt a more abstract approach, starting from the Pentagon identity.

### 2.1 Case of the modular kernel

**Explicit shift equations**

The derivation of the shift equations for the modular kernel $\mathbf{M}^{(b)}_{P_s, P_t}[P_0]$ is done explicitly in [18, Appendix B]. The resulting equations are valid for any $c \in \mathbb{C}$. In our notations and

normalization, the equations read

$$\sum_{\pm} A_b(\pm P_s, P_0) e^{\pm \frac{b}{2} \partial_{P_s}} \mathbf{M}^{(b)}_{P_s, P_t}[P_0] = 2\cos(2\pi b P_t) \mathbf{M}^{(b)}_{P_s, P_t}[P_0], \tag{2.1a}$$

$$\sum_{\pm} e^{\pm \frac{b}{2} \partial_{P_t}} A_b(\mp P_t, P_0) \mathbf{M}^{(b)}_{P_s, P_t}[P_0] = 2\cos(2\pi b P_s) \mathbf{M}^{(b)}_{P_s, P_t}[P_0], \tag{2.1b}$$

$$\sum_{\pm} e^{-b \partial_{P_0}} E_b(\pm P_s, P_t, P_0) e^{\pm \frac{b}{2} \partial_{P_s}} \mathbf{M}^{(b)}_{P_s, P_t}[P_0] = \mathbf{M}^{(b)}_{P_s, P_t}[P_0], \tag{2.1c}$$

$$\sum_{\pm} e^{\pm \frac{b}{2} \partial_{P_t}} E_b(\mp P_t, P_s, -P_0) e^{-b \partial_{P_0}} \mathbf{M}^{(b)}_{P_s, P_t}[P_0] = \mathbf{M}^{(b)}_{P_s, P_t}[P_0], \tag{2.1d}$$

where the coefficients are combinations of Gamma functions,

$$A_b(P_s, P_0) = \frac{\Gamma(2bP_s)\Gamma(1+b^2+2bP_s)}{\prod_{\pm} \Gamma(\frac{1+b^2}{2} \pm bP_0 + 2bP_s)}, \tag{2.2}$$

$$E_b(P_s, P_t, P_0) = \frac{1}{2\pi} \Gamma(2bP_s)\Gamma(1+b^2+2bP_s) \frac{\prod_{\pm} \Gamma(\frac{1}{2} - \frac{b^2}{2} - bP_0 \pm 2bP_t)}{\prod_{\pm} \Gamma(\frac{1}{2} \pm \frac{b^2}{2} - bP_0 + 2bP_s)}. \tag{2.3}$$

The fourth equation (2.1d) is not found in [18], and it is in principle redundant, because the dependence on $P_0$ is already taken care of by the third equation (2.1c). We include the fourth equation for completeness and for later convenience. We have derived it from the third equation using the fact that the modular transformation squares to identity, $\int dP_t \quad \mathbf{M}^{(b)}_{P_s, P_t}[P_0] \mathbf{M}^{(b)}_{P_t, P'_t}[P_0] \quad = \quad \delta(P_s - P'_t)$, and using reflection invariance $\mathbf{M}^{(b)}_{P_s, P_t}[P_0] = \mathbf{M}^{(b)}_{P_s, P_t}[-P_0]$.

**Behaviour under Virasoro-Wick rotation**

We will now show that the shift equations are invariant under the transformation

$$\boxed{\mathfrak{R}\mathbf{M}^{(b)}_{P_s, P_t}[P_0] := \frac{P_t}{P_s} \mathbf{M}^{(ib)}_{iP_t, iP_s}[iP_0].} \tag{2.4}$$

To do this, we start with the second shift equation (2.1b), where we Virasoro–Wick-rotate the variables, rename $P_s \leftrightarrow P_t$, and move the shift operator $e^{\mp \frac{b}{2} \partial_{P_t}}$ past the coefficient function $A_b$:

$$\sum_{\pm} A_{ib}\left(\mp iP_s - i\frac{b}{2}, iP_0\right) e^{\pm \frac{b}{2} \partial_{P_s}} \mathbf{M}^{(ib)}_{iP_t, iP_s}[iP_0] = 2\cos(2\pi b P_t) \mathbf{M}^{(ib)}_{iP_t, iP_s}[iP_0]. \tag{2.5}$$

Then we notice that the coefficient function $A_b$ (2.2) obeys the identity

$$A_{ib}\left(-iP_s - i\frac{b}{2}, iP_0\right) = \frac{P_s}{P_s + \frac{b}{2}} A_b(P_s, P_0). \tag{2.6}$$

It immediately follows that $\mathfrak{R}\mathbf{M}$ obeys the first shift equation (2.1a). Similarly, the second shift equation could be deduce from the first, using the same identity for $A_b$.

Then let us rewrite the fourth shift equation (2.1d) and again Virasoro–Wick-rotate the variables, rename $P_s \leftrightarrow P_t$, and move the shift operators $e^{\mp \frac{b}{2} \partial_{P_t}}$ and $e^{-b \partial_{P_0}}$ past the coefficient function $E_b$:

$$\sum_{\pm} e^{-b \partial_{P_0}} E_{ib}\left(\mp iP_s - i\frac{b}{2}, iP_t, -iP_0 - ib\right) e^{\pm \frac{b}{2} \partial_{P_s}} \mathbf{M}^{(ib)}_{iP_t, iP_s}[iP_0] = \mathbf{M}^{(ib)}_{iP_t, iP_s}[iP_0]. \tag{2.7}$$

Then we notice that the coefficient function $E_b$ (2.3) obeys the identity

$$E_{ib}\left(-iP_s - i\tfrac{b}{2}, iP_t, -iP_0 - ib\right) = \frac{P_s}{P_s + \frac{b}{2}} E_b(P_s, P_t, P_0). \tag{2.8}$$

It immediately follows that $\Re\mathbf{M}$ obeys the third shift equation (2.1c). Similarly, the fourth shift equation could be deduced from the third, using the same identity for $E_b$.

**Solution for $c \in \mathbb{C} \backslash (-\infty, 1]$**

The modular kernel is explicitly known as an integral expression [24, 25]:

$$\mathbf{M}^{(b)}_{P_s, P_t}[P_0] = \frac{\sqrt{2}}{S_b(\frac{Q}{2} + P_0)} \prod_{\pm} \frac{\Gamma_b(Q \pm 2P_s)}{\Gamma_b(\pm 2P_t)} \frac{\Gamma_b(\frac{Q}{2} - P_0 \pm 2P_t)}{\Gamma_b(\frac{Q}{2} - P_0 \pm 2P_s)} \int_{i\mathbb{R}} \frac{du}{i} e^{4\pi i P_s u} \prod_{\pm, \pm} S_b\left(\frac{Q}{4} + \frac{P_0}{2} \pm u \pm P_t\right). \tag{2.9}$$

Here we use the standard notation $Q = b + b^{-1}$, as well as Barnes' double Gamma function $\Gamma_b(x)$, and the double Sine function $S_b(x) = \frac{\Gamma_b(x)}{\Gamma_b(Q-x)}$.

It is in principle straightforward to check that the integral expression is indeed a solution of the shift equations. The idea is that the shift equations for the integral follow from the shift equations for the integrand. In practice, if we write the integral factor as $\int_{i\mathbb{R}} du\, \varphi(u)$, then the integrand $\varphi(u)$ obeys equations of the type $\varphi(u)k_1(u) = \varphi(u+b)k_2(u)$ where $k_1(u), k_2(u)$ are trigonometric functions, as a result of the behaviour of the double Sine function $\frac{S_b(x+b)}{S_b(x)} = 2\sin(\pi b x)$. This leads to $\int_{i\mathbb{R}} du\, \varphi(u)(k_1(u) - k_2(u-b)) = 0$. (In the shift $u \to u - b$, no pole of $\varphi$ crosses the contour of integration.) As a function of $u$, the factor $k_1(u) - k_2(u - b)$ is a linear combination of the three terms $1, e^{2\pi i b u}, e^{-2\pi i b u} = 1, \frac{e^{\frac{b}{2}\partial_{P_s}}\varphi(u)}{\varphi(u)}, \frac{e^{-\frac{b}{2}\partial_{P_s}}\varphi(u)}{\varphi(u)}$. This results in a shift equation for $\int_{i\mathbb{R}} du\, \varphi(u)$, which is equivalent to the first shift equation (2.1a) for the modular kernel.

## 2.2 Case of the fusion kernel

**Shift equations from the Pentagon identity**

The fusion kernel was originally derived by Ponsot and Teschner by solving shift equations [10]. Let us rederive these equations from the Pentagon identity [4]

$$\int_{i\mathbb{R}} \frac{dP_{23}}{i} \mathbf{F}^{(b)}_{P_{12}, P_{23}} \begin{bmatrix} P_2 & P_3 \\ P_1 & P_{45} \end{bmatrix} \mathbf{F}^{(b)}_{P_{45}, P_{51}} \begin{bmatrix} P_{23} & P_4 \\ P_1 & P_5 \end{bmatrix} \mathbf{F}^{(b)}_{P_{23}, P_{34}} \begin{bmatrix} P_3 & P_4 \\ P_2 & P_{15} \end{bmatrix} = \mathbf{F}^{(b)}_{P_{45}, P_{34}} \begin{bmatrix} P_3 & P_4 \\ P_{12} & P_5 \end{bmatrix} \mathbf{F}^{(b)}_{P_{12}, P_{51}} \begin{bmatrix} P_2 & P_{34} \\ P_1 & P_5 \end{bmatrix}. \tag{2.10}$$

Let us focus on the two special cases where the field 2 or 3 is a degenerate field $V_{\langle 2,1 \rangle}$ with a vanishing null vector at level two. Due to the fusion rule $V_{\langle 2,1 \rangle} V_P \sim \sum_{\pm} V_{P \pm \frac{b}{2}}$, the integral over the momentum $P_{23}$ is replaced with a sum of two terms, and we obtain shift equations. Renaming the six momentums that remain independent $P_1, P_2, P_3, P_4, P_s, P_t$, and introducing the signs $\eta, \nu \in \{+, -\}$, we obtain

$$\sum_{\epsilon = \pm} \mathbf{F}^{(b)}_{P_s, P_t} \begin{bmatrix} P_2 + \epsilon \frac{b}{2} & P_3 \\ P_1 & P_4 \end{bmatrix} \mathbf{f}^{(b)}_{\eta, \epsilon} \begin{bmatrix} P_1 & P_s \\ \langle 2,1 \rangle & P_2 \end{bmatrix} \mathbf{f}^{(b)}_{\epsilon, \nu} \begin{bmatrix} P_2 & P_3 \\ \langle 2,1 \rangle & P_t \end{bmatrix} = \mathbf{F}^{(b)}_{P_s, P_t + \nu \frac{b}{2}} \begin{bmatrix} P_2 & P_3 \\ P_1 + \eta \frac{b}{2} & P_4 \end{bmatrix} \mathbf{f}^{(b)}_{\eta, -\nu} \begin{bmatrix} P_1 & P_4 \\ \langle 2,1 \rangle & P_t + \nu \frac{b}{2} \end{bmatrix}, \tag{2.11}$$

$$\sum_{\epsilon = \pm} \mathbf{F}^{(b)}_{P_s, P_t} \begin{bmatrix} P_2 + \epsilon \frac{b}{2} & P_3 \\ P_1 & P_4 \end{bmatrix} \mathbf{f}^{(b)}_{\eta, \epsilon} \begin{bmatrix} P_s & P_1 \\ \langle 2,1 \rangle & P_2 \end{bmatrix} \mathbf{f}^{(b)}_{\epsilon, \nu} \begin{bmatrix} P_2 & P_t \\ \langle 2,1 \rangle & P_3 \end{bmatrix} = \mathbf{F}^{(b)}_{P_s + \eta \frac{b}{2}, P_t} \begin{bmatrix} P_2 & P_3 + \nu \frac{b}{2} \\ P_1 & P_4 \end{bmatrix} \mathbf{f}^{(b)}_{-\eta, \nu} \begin{bmatrix} P_s + \eta \frac{b}{2} & P_4 \\ \langle 2,1 \rangle & P_3 \end{bmatrix}, \tag{2.12}$$

where we introduce degenerate fusion kernels, which are monodromy matrices of hypergeometric Belavin–Polyakov–Zamolodchikov differential equations for four-point functions of the type $\left\langle V_{\langle 2,1 \rangle} V_{P_1} V_{P_2} V_{P_3} \right\rangle$ [4],

$$\mathbf{f}^{(b)}_{\epsilon, \eta} \begin{bmatrix} P_1 & P_2 \\ \langle 2,1 \rangle & P_3 \end{bmatrix} := \mathbf{F}^{(b)}_{P_1 + \epsilon \frac{b}{2}, P_3 + \eta \frac{b}{2}} \begin{bmatrix} P_1 & P_2 \\ \langle 2,1 \rangle & P_3 \end{bmatrix} = \frac{\Gamma(1 - 2b\epsilon P_1)\Gamma(2b\eta P_3)}{\prod_{\pm} \Gamma(\frac{1}{2} - b\epsilon P_1 \pm bP_2 + b\eta P_3)}. \tag{2.13}$$

In these formulas, by $\langle 2,1 \rangle$ we really mean a degenerate representation with a vanishing null vector at level 2. We do not rely on the fact that the corresponding fusion kernel can be obtained as a limit $P \to P_{(2,1)}$ of a fusion kernel for Verma modules. In particular, the fusion rules of the corresponding primary field $V_{\langle 2,1 \rangle}$ are obeyed by definition.

**Behaviour under Virasoro–Wick rotation**

While the hypergeometric equation does not behave particularly nicely under Virasoro–Wick rotation, its monodromy matrix does:

$$\mathbf{f}^{(ib)}_{\epsilon,\eta} \begin{bmatrix} iP_1 & iP_2 \\ \langle 2,1 \rangle & iP_3 \end{bmatrix} = -\epsilon \eta \frac{P_1}{P_3} \mathbf{f}^{(b)}_{\eta,\epsilon} \begin{bmatrix} P_3 & P_2 \\ \langle 2,1 \rangle & P_1 \end{bmatrix}. \tag{2.14}$$

Let us use this identity for simplifying the the first shift equation (2.11) with Virasoro–Wick-rotated variables:

$$\sum_{\epsilon=\pm} \frac{1}{P_t} \mathbf{F}^{(ib)}_{iP_s,iP_t} \begin{bmatrix} iP_2 + \epsilon \frac{ib}{2} & iP_3 \\ iP_1 & iP_4 \end{bmatrix} \mathbf{f}^{(b)}_{\nu,\epsilon} \begin{bmatrix} P_t & P_3 \\ \langle 2,1 \rangle & P_2 \end{bmatrix} \mathbf{f}^{(b)}_{\epsilon,\eta} \begin{bmatrix} P_2 & P_s \\ \langle 2,1 \rangle & P_1 \end{bmatrix} = \frac{1}{P_t + \nu \frac{b}{2}} \mathbf{F}^{(ib)}_{iP_s,iP_t+\nu\frac{ib}{2}} \begin{bmatrix} iP_2 & iP_3 \\ iP_1 + \eta \frac{ib}{2} & iP_4 \end{bmatrix} \mathbf{f}^{(b)}_{-\nu,\eta} \begin{bmatrix} P_t + \nu \frac{b}{2} & P_4 \\ \langle 2,1 \rangle & P_1 \end{bmatrix}. \tag{2.15}$$

We compare this with the second shift equation (2.12), where we do the renamings $P_s \leftrightarrow P_t$, $P_1 \leftrightarrow P_3$ and $\nu \leftrightarrow \eta$:

$$\sum_{\epsilon=\pm} \mathbf{F}^{(b)}_{P_t,P_s} \begin{bmatrix} P_2 + \epsilon \frac{b}{2} & P_1 \\ P_3 & P_4 \end{bmatrix} \mathbf{f}^{(b)}_{\nu,\epsilon} \begin{bmatrix} P_t & P_3 \\ \langle 2,1 \rangle & P_2 \end{bmatrix} \mathbf{f}^{(b)}_{\epsilon,\eta} \begin{bmatrix} P_2 & P_s \\ \langle 2,1 \rangle & P_1 \end{bmatrix} = \mathbf{F}^{(b)}_{P_t+\nu\frac{b}{2},P_s} \begin{bmatrix} P_2 & P_1 + \eta \frac{b}{2} \\ P_3 & P_4 \end{bmatrix} \mathbf{f}^{(b)}_{-\nu,\eta} \begin{bmatrix} P_t + \nu \frac{b}{2} & P_4 \\ \langle 2,1 \rangle & P_1 \end{bmatrix}. \tag{2.16}$$

The degenerate fusion kernels are now the same as in the Virasoro–Wick-rotated equation (2.15). This shows that the rotation $\mathbf{F} \to \mathfrak{R}\mathbf{F}$ (1.4) is compatible with the shift equations, in the sense that $\mathbf{F}$ obeys the second shift equation if and only if $\mathfrak{R}\mathbf{F}$ obeys the first one — and vice versa.

One should resist the temptation of trying to deduce the Pentagon identity (2.10) at $c$ from the Pentagon identity at $26 - c$. While this works at the level of the integrand and right-hand side, the integration contours at $c$ and $26 - c$ are not related by Virasoro–Wick rotation.

**Teschner–Vartanov formula for $c \in \mathbb{C} \backslash (-\infty, 1]$**

Let us write the known solution of the shift equations. We choose the Teschner–Vartanov formula [11] rather than the earlier Ponsot–Teschner formula [10], in order to make tetrahedral symmetry manifest. This allows us to write a relatively compact expression for the fusion kernel using the tetrahedral notations (1.6):

$$\mathbf{F}^{(b)}_{P_s,P_t} \begin{bmatrix} P_2 & P_3 \\ P_1 & P_4 \end{bmatrix} = \frac{\Gamma_b(Q \pm 2P_s)}{2\Gamma_b(\pm 2P_t)} \prod_{f \in F} \prod_{\substack{\sigma \in \mathbb{Z}_2^f \\ \sigma_f = \eta_t(f)}} \Gamma_b \left( \frac{Q}{2} + \sum_{i \in f} \sigma_i P_i \right)^{\sigma_f} \int_{\frac{Q}{2}+i\mathbb{R}} \frac{du}{i} \prod_{\substack{\sigma \in \mathbb{Z}_2^E \\ \sigma_V = 1}} S_b \left( u + \frac{Q\sigma_E}{4} + \frac{1}{2} \sum_{i \in E} \sigma_i P_i \right)^{-\sigma_E}, \tag{2.17}$$

where we use the same special functions as for the modular kernel (2.9). Apart from notational differences, our formula differs from [11] because of our use of the natural normalization (1.10) for conformal blocks.

# 3 Interpretation of the Virasoro–Wick rotation

In this section we will try to make sense of the Virasoro–Wick rotation, when applied to the known fusion and modular kernels. We will first discuss the analytic properties of the kernels in the regime $c < 1$, where they are not known in closed form. Then we will apply the rotation

to Liouville theory, and in particular to the crossing and modular equations that relate the kernels with the structure constants.

We will now reserve the notations $\mathbf{F}, \mathbf{M}$ for the known fusion and modular kernels, which are unique solutions of the shift equations for $c \in \mathbb{C}\backslash(-\infty, 1]$. We will write $\mathfrak{R}\mathbf{F}, \mathfrak{R}\mathbf{M}$ their images under the rotations (1.4) and (2.4). We will call $\hat{\mathbf{F}}, \hat{\mathbf{M}}$ the kernels for $c \leq 1$, which are still unknown (except $\hat{\mathbf{F}}$ for $c = 1$), and $\mathfrak{R}\hat{\mathbf{F}}, \mathfrak{R}\hat{\mathbf{M}}$ their images under the rotation. Let us summarize the domains of definition of these kernels:

| Kernels | $c \leq 1$ | $c \in \mathbb{C}\backslash(-\infty, 1] \cup [25, \infty)$ | $c \geq 25$ |
|---|---|---|---|
| $\mathbf{F}, \mathbf{M}$ | | 🟩 | 🟩 |
| $\mathfrak{R}\mathbf{F}, \mathfrak{R}\mathbf{M}$ | 🟩 | 🟩 | |
| $\hat{\mathbf{F}}, \hat{\mathbf{M}}$ | 🟩 | | |
| $\mathfrak{R}\hat{\mathbf{F}}, \mathfrak{R}\hat{\mathbf{M}}$ | | | 🟩 |

(3.1)

## 3.1 How smooth are the fusion and modular kernels for $c < 1$?

**Argument from uniqueness**

For any $b \in \mathbb{R}$ such that $b^2 \notin \mathbb{Q}$, any continuous function on $\mathbb{R}$ that obeys $f(P + b) = f(P + b^{-1}) = f(P)$ is constant. Under the assumption that it is meromorphic, the function $f$ is furthermore constant over the complex $P$-plane. Under the assumption that $f$ depends continuously on $b$, we conclude that it is $P$-independent for any $b \in \mathbb{R}$. And if $f$ is a meromorphic function of $b \in \mathcal{B}$ (for $\mathcal{B} \subset \mathbb{C}$ an open domain), then it is $P$-independent for any $b \in \mathcal{B}$.

These statements are the basis for the bootstrap derivation of the Liouville three-point structure constant [26]. A similar reasoning has been applied to the Virasoro fusion kernel, originally in [27], with the proof of uniqueness completed in [16]. The outcome is that for $c \in \mathbb{C}\backslash(-\infty, 1]$, the Ponsot–Teschner kernel $\mathbf{F}$ is the unique solution of the shift equations that is meromorphic in $b$ and in the momentums. And there is every reason to believe that the modular kernel $\mathbf{M}$ is also the unique solution of its own shift equations.

By Virasoro–Wick rotation, it follows that $\mathfrak{R}\mathbf{F}$ and $\mathfrak{R}\mathbf{M}$ are the unique solutions for $c \in \mathbb{C}\backslash[25, \infty)$, under the same meromorphicity assumptions. However, these solutions cannot be kernels, because they are odd in $P_s$ and $P_t$. Since the conformal blocks are even, the integral in the fusion transformation (1.3) must vanish. As we have checked numerically, this conclusion holds even for $c \leq 1$, in which case the conformal blocks' poles at degenerate momentums $\left(P_{(r,s)}\right)_{r,s \in \mathbb{N}^*}$ (1.5) sit on the integration line, and force us to move that line as $i\mathbb{R} \to i\mathbb{R} + \Lambda$ with $\Lambda \in \mathbb{R}^*$ [16, 28]. And even if the integral over $P_t$ did not vanish, the result could not be an $s$-channel block, due to parity in $P_s$.

For $c \in \mathbb{C}\backslash(-\infty, 1] \cup [25, \infty)$, the existence of two solutions of the shift equations is not a problem. These solutions are unique only if we assume that they are meromorphic on $c \in \mathbb{C}\backslash(-\infty, 1]$ and $c \in \mathbb{C}\backslash[25, \infty)$ respectively. The core of the uniqueness argument is indeed the existence of two shifts $b$ and $b^{-1}$ that are aligned in the complex plane, which occurs for $c \leq 1$ or $c \geq 25$:

$$
\begin{array}{cccc}
\overset{i}{\underset{0 \quad 1}{\llcorner}}\,, & \bigg\updownarrow\,, & \diagup\!\!\!\diagdown\,, & \longrightarrow\,, \\
& b \in i\mathbb{R}, & b^2 \notin \mathbb{R}, & b \in \mathbb{R}, \\
& c \leq 1, & c \in \mathbb{C}, & c \geq 25.
\end{array}
$$

(3.2)

Solutions on half-line are then extended to the complex plane by analytic continuation, using the fact that they are built from the double Gamma function $\Gamma_b(x)$, which is meromorphic for $\Im b > 0$. We therefore conclude that for $c \in \mathbb{C}\backslash(-\infty, 1]\cup[25, \infty)$, the fusion kernel is $\mathbf{F}$, while $\Re\mathbf{F}$ is unphysical.

For $c \leq 1$ however, $\mathbf{F}$ does not exist, while $\Re\mathbf{F}$ is still odd, and therefore differs from the fusion kernel $\hat{\mathbf{F}}$. Since $\Re\mathbf{F}$ is meromorphic in momentums, $\hat{\mathbf{F}}$ cannot be meromorphic, by the uniqueness argument of [16]. (Actually, for $b^2 \notin \mathbb{Q}$, continuity might well suffice for uniqueness.)

### Limit from degenerate cases

For $c \leq 1$, the degenerate momentums $P_{(r,s)}$ (1.5) are dense in the imaginary axis, and it is therefore possible to obtain generic blocks and kernels as limits of degenerate blocks and kernels. Degenerate fusion kernels such as $\mathbf{F}^{(b)}_{P_2+P_{(m,n)},P_t}\begin{bmatrix} P_2 & P_3 \\ P_{(r,s)} & P_4 \end{bmatrix}$ with $r,s \in \mathbb{N}^*$ and $|m| \in r-1-2\mathbb{N}$ and $|n| \in s-1-2\mathbb{N}$ can be deduced from the known integral formula (2.17), where the integral reduces to a discrete sum, and the ratios of double Gamma and double Sine functions reduce to products of Gamma functions. While the original integral formula is only valid for $c \in \mathbb{C}\backslash(-\infty, 1]$, the resulting expression for the degenerate kernels is therefore valid for all $c \in \mathbb{C}$.

For $c \leq 1$, it only remains to take an appropriate $r,s,m,n \to \infty$ limit, such that we recover generic values of the momentums. Taking this limit is an interesting technical challenge. A similar limit was performed in the case of three-point structure constants of the non-rational limit of D-series minimal models [29]. In that case, assuming $b^2 \notin \mathbb{Q}$, it was found that the limit structure constants are not smooth functions of momentums, but distributions. This suggests that the fusion kernel $\hat{\mathbf{F}}$ is also a distribution: not only by analogy, but also because the D-series structure constants can be written in terms of the fusion kernel [30]. If the structure constants become distributions in the limit, then the fusion kernel should become a distribution as well.

### Conjecture

We therefore conjecture that for $c < 1$ with $b^2 \notin \mathbb{Q}$, the fusion and modular kernels are not smooth functions of the momentums, but distributions. In terms of the fusion transformation of four-point conformal blocks, this means that instead of a single integral (1.3) with a well-defined kernel, we expect a series of the type

$$\mathcal{F}^{(b),s-\text{channel}}_{P_s} = \sum_{k=1}^{\infty} \int_{i\mathbb{R}+\Lambda} \frac{dP_t}{i} \, \hat{\mathbf{F}}^{(b),k}_{P_s,P_t} \mathcal{F}^{(b),t-\text{channel}}_{P_t}, \tag{3.3}$$

with an infinite family of kernels $\hat{\mathbf{F}}^{(b),k}_{P_s,P_t}$ which may well be meromorphic functions in the momentums. The sum of these kernels would be divergent, but it would become convergent after integration against the $t$-channel conformal blocks. A divergent series expression of this type is what emerges for the three-point structure constants in [29], and it is what we would expect if we took a limit from degenerate cases.

### Series representation

Recently, Roussillon has proposed a series representation of the fusion and modular kernels, which is valid for $b^2 \notin \mathbb{Q}$ [31]. If true, this would in particular determine the fusion kernel $\hat{\mathbf{F}}$ for almost all $c < 1$. We will now argue that these results are consistent with our conjecture.

If the series for $\hat{\mathbf{F}}$ was divergent, it would directly fulfill our expectation (3.3). However, the series looks convergent for $\Re P_t > \Lambda$ for some $\Lambda \geq 0$, under the mild restriction that $b^2$ is

not a Liouville number. Proving convergence is not easy, and we proceed under the assumption that the series is absolutely convergent, leading to a kernel that is meromorphic.

This looks problematic not only for our conjecture, but also for the uniqueness of the kernel. However, uniqueness would only apply if the kernel was meromorphic over $P_t \in \mathbb{C}$, so there is no contradiction [16]. Our second argument that the kernel is not smooth was based on taking a limit from degenerate cases. This limit produces a kernel that is even as a function of $P_t$, whereas the series representation is undefined for $\Re P_t \leq 0$. Again there is no contradiction: summing the series produces a kernel that obeys the fusion relation (1.3), but differs from what we would call the physical kernel, a distribution that is even in $P_t$. In this respect, the kernel is similar to the diagonal three-point structure constant of [29], which is given by the divergent sum

$$C^D_{P_1,P_2,P_3} \propto 1 + 2 \sum_{n=1}^{\infty} (-)^n \frac{\prod_{i=1}^3 \cos(2\pi n b P_i)}{\cos(\pi n b^2)}, \qquad (3.4)$$

where we omit a smooth prefactor. However, the modified structure constant,

$$\widetilde{C}^D_{P_1,P_2,P_3} \propto 1 + 2 \sum_{n=1}^{\infty} (-)^n \frac{\cos(2\pi n b P_1)\cos(2\pi n b P_2)e^{2\pi i n b P_3}}{\cos(\pi n b^2)}, \qquad (3.5)$$

converges in a large domain of values of $P_3$, at the expense of breaking the invariances under $P_3 \to -P_3$ and $P_3 \leftrightarrow P_{1,2}$. This symmetry breaking is not a problem so long we integrate over $P_3$ in the context of a correlation function.

## 3.2   Application to Liouville theory

Since Liouville theory is diagonal and has a continuous spectrum, its structure constants are closely related to the Virasoro fusion kernel. The bulk three-point structure constant is a special case of the fusion kernel, while the boundary three-point structure constant coincides with the fusion kernel up to simple prefactors [27]. Here we will review the crossing and modular invariance equations of Liouville theory, which involve the bulk three-point structure constant as well as the fusion and modular kernels. Then we will see what happens to these equations when we apply the Virasoro–Wick rotation.

**Crossing and modular invariance**

For $c \in \mathbb{C} \backslash (-\infty, 1]$, there is a field normalization such that the two- and three-point structure constants of Liouville theory are [4]

$$B_P^{(b)} = \prod_{\pm} \Gamma_b(\pm 2P)\Gamma_b(Q \pm 2P), \qquad C^{(b)}_{P_1,P_2,P_3} = \prod_{\pm,\pm,\pm} \Gamma_b \left( \tfrac{Q}{2} \pm P_1 \pm P_2 \pm P_3 \right). \qquad (3.6)$$

For $c \leq 1$, the structure constants may be written as

$$\hat{B}_P^{(b)} = \frac{1}{4P^2 B_{iP}^{(ib)}}, \qquad \hat{C}^{(b)}_{P_1,P_2,P_3} = \frac{1}{C^{(ib)}_{iP_1,iP_2,iP_3}}. \qquad (3.7)$$

These relations between the structure constants for $c \leq 1$ and $c \geq 25$ give rise to great simplifications when coupling theories with central charges $c$ and $26-c$ in order to build a two-dimensional gravity theory [14] or a string theory [32]. Actually, the Virasoro–Wick rotation unexpectedly shows up as a symmetry of the Virasoro minimal string [32, Section 4.5], which is built from two coupled Liouville theories: our analysis might be useful to make sense of this symmetry.

Since Liouville theory is diagonal, crossing symmetry of the sphere four-point function may be rewritten as [4]

$$\frac{C^{(b)}_{P_1,P_2,P_s} C^{(b)}_{P_s,P_3,P_4}}{B^{(b)}_{P_s}} \mathbf{F}^{(b)}_{P_s,P_t} \begin{bmatrix} P_2 & P_3 \\ P_1 & P_4 \end{bmatrix} = \frac{C^{(b)}_{P_2,P_3,P_t} C^{(b)}_{P_1,P_4,P_t}}{B^{(b)}_{P_t}} \mathbf{F}^{(b)}_{P_t,P_s} \begin{bmatrix} P_2 & P_1 \\ P_3 & P_4 \end{bmatrix}. \tag{3.8}$$

Similarly, modular invariance of the torus one-point function may be rewritten as

$$\frac{C^{(b)}_{P_s,P_s,P_0}}{B^{(b)}_{P_s}} \mathbf{M}^{(b)}_{P_s,P_t}[P_0] = \frac{C^{(b)}_{P_t,P_t,P_0}}{B^{(b)}_{P_t}} \mathbf{M}^{(b)}_{P_t,P_s}[P_0]. \tag{3.9}$$

In this formulation, crossing symmetry and modular invariance of Liouville theory can be deduced from the properties of the fusion and modular kernels, after writing the structure constants in terms of the fusion kernel. This provides the algebraic half of a proof of consistency of Liouville theory. The analytic half would be to show that the decompositions of correlation functions into conformal blocks actually converge.

**Virasoro–Wick rotation**

For $c \leq 1$, since Liouville four-point functions are crossing-symmetric [28], the fusion kernel $\hat{\mathbf{F}}$ and the Liouville structure constants (3.7) should obey the same equation (3.8) as for $c \in \mathbb{C}\backslash(-\infty, 1]$,

$$\frac{\hat{C}^{(b)}_{P_1,P_2,P_s} \hat{C}^{(b)}_{P_s,P_3,P_4}}{\hat{B}^{(b)}_{P_s}} \hat{\mathbf{F}}^{(b)}_{P_s,P_t} \begin{bmatrix} P_2 & P_3 \\ P_1 & P_4 \end{bmatrix} = \frac{\hat{C}^{(b)}_{P_2,P_3,P_t} \hat{C}^{(b)}_{P_1,P_4,P_t}}{\hat{B}^{(b)}_{P_t}} \hat{\mathbf{F}}^{(b)}_{P_t,P_s} \begin{bmatrix} P_2 & P_1 \\ P_3 & P_4 \end{bmatrix}. \tag{3.10}$$

Since we do not know the kernel $\hat{\mathbf{F}}$, we cannot check this equation. However, if we apply the Virasoro–Wick rotation to Eq. (3.8), we obtain a relation that involves the rotated fusion kernel $\mathfrak{R}\mathbf{F}$ (1.4),

$$\frac{\hat{C}^{(b)}_{P_1,P_2,P_s} \hat{C}^{(b)}_{P_s,P_3,P_4}}{\hat{B}^{(b)}_{P_s}} \mathfrak{R}\mathbf{F}^{(b)}_{P_s,P_t} \begin{bmatrix} P_2 & P_3 \\ P_1 & P_4 \end{bmatrix} = \frac{\hat{C}^{(b)}_{P_2,P_3,P_t} \hat{C}^{(b)}_{P_1,P_4,P_t}}{\hat{B}^{(b)}_{P_t}} \mathfrak{R}\mathbf{F}^{(b)}_{P_t,P_s} \begin{bmatrix} P_2 & P_1 \\ P_3 & P_4 \end{bmatrix}. \tag{3.11}$$

Therefore, the Virasoro–Wick rotation yields an equation for $\mathfrak{R}\mathbf{F}$ that is identical to the equation for $\hat{\mathbf{F}}$. Technically, this is because:

- In the crossing symmetry equation (3.10), the left-hand side kernel is related to the right-hand side kernel by the same permutation of momentums that appears in the Virasoro–Wick rotation for the fusion kernel. The Virasoro–Wick rotation therefore exchanges these kernels, and this compensates the fact that it inverses the structure constants.

- The simple prefactor $\frac{P_t}{P_s}$ of the rotated fusion kernel cancels the prefactor $P^2$ of the two-point structure constant.

By the same mechanism, the rotated modular kernel $\mathfrak{R}\mathbf{M}$ (2.4), together with the $c \leq 1$ Liouville structure constants, obey a relation that is identical to the modular invariance relation (3.9):

$$\frac{\hat{C}^{(b)}_{P_s,P_s,P_0}}{\hat{B}^{(b)}_{P_s}} \mathfrak{R}\mathbf{M}^{(b)}_{P_s,P_t}[P_0] = \frac{\hat{C}^{(b)}_{P_t,P_t,P_0}}{\hat{B}^{(b)}_{P_t}} \mathfrak{R}\mathbf{M}^{(b)}_{P_t,P_s}[P_0]. \tag{3.12}$$

Therefore, everything works as if we were deducing crossing symmetry of Liouville theory with $c \leq 1$ from Liouville theory with $c \geq 25$, except of course that $\mathfrak{R}\mathbf{F}$ and $\mathfrak{R}\mathbf{M}$ are not the actual fusion and modular kernels for $c \leq 1$, but unphysical solutions of the shift equations. While crossing symmetry equations are very strong constraints on structure constants, they are weak constraints on fusion kernels, so it is not too surprising that two different kernels obey these constraints.

# 4 The $c = 25$ fusion kernel

Let us start with a special case at $c = 1$ and $c = 25$ where all relevant kernels have very simple expressions [33]. This special case is defined by $\Delta_i = \frac{1}{16}$ for $c = 1$, and $\Delta_i = \frac{15}{16}$ for $c = 25$, with $i = 1, 2, 3, 4$. We define the following kernels:

| $c = 1$ | $c = 25$ |
|---|---|
| $\hat{\mathbf{F}}^{\pm}_{P_s, P_t} = 16^{P_t^2 - P_s^2} e^{\mp 2\pi i P_s P_t}$ | $\mathbf{F}^{\pm}_{P_s, P_t} = \pm i \frac{P_t}{P_s} 16^{P_t^2 - P_s^2} e^{\pm 2\pi i P_s P_t}$ |
| $\hat{\mathbf{F}}_{P_s, P_t} = 16^{P_t^2 - P_s^2} \cos(2\pi P_s P_t)$ | $\mathbf{F}_{P_s, P_t} = -\frac{P_t}{P_s} 16^{P_t^2 - P_s^2} \sin(2\pi P_s P_t)$ |
| $\mathfrak{R}\mathbf{F}_{P_s, P_t} = 16^{P_t^2 - P_s^2} \sin(2\pi P_s P_t)$ | $\mathfrak{R}\hat{\mathbf{F}}_{P_s, P_t} = \frac{P_t}{P_s} 16^{P_t^2 - P_s^2} \cos(2\pi P_s P_t)$ |

$$(4.1)$$

Here $\mathbf{F}^{\pm}, \mathbf{F}, \hat{\mathbf{F}}^{\pm}, \hat{\mathbf{F}}$ are all kernels that obey the fusion relation (1.3) for this special case, with $\mathbf{F} = \frac{1}{2} \sum_{\pm} \mathbf{F}^{\pm}$ and $\hat{\mathbf{F}} = \frac{1}{2} \sum_{\pm} \hat{\mathbf{F}}^{\pm}$. On the other hand, the odd, unphysical combinations $\mathfrak{R}\mathbf{F}$ and $\mathfrak{R}\hat{\mathbf{F}}$ vanish when integrated against conformal blocks. Under the Virasoro–Wick rotation (1.4), these kernels behave as

$$\mathfrak{R}\hat{\mathbf{F}}^{\pm} = \mp i \mathbf{F}^{\pm}. \tag{4.2}$$

In this section we will see that this picture is valid at $c = 1$ and $c = 25$ not only in our special case, but also for arbitrary values of $\Delta_i$. However, for generic values, it turns out that only the kernels $\mathbf{F}$ and $\mathfrak{R}\mathbf{F}$ are meromorphic in $P_t, P_s$: all the rest have branch cuts.

## 4.1 The $c = 1$ fusion kernel revisited

**Tetrahedral notation**

Thanks to the relation between $c = 1$ conformal blocks and tau functions of the Painlevé VI equation [34], it is possible to write the $c = 1$ fusion kernel in terms of the connection coefficient of that nonlinear differential equation [12]. In order to make the symmetries more manifest, let us write the resulting expression in our tetrahedral notation (1.6). We also define the quantities $\hat{d}$ and $\hat{\omega}_{\pm}$ by Virasoro–Wick-rotating the momentums as $P_i \to i P_i$ in Eqs. (1.8) and (1.9). This leads to the two kernels

$$\hat{\mathbf{F}}^{\epsilon = \pm}_{P_s P_t} \begin{bmatrix} P_2 & P_3 \\ P_1 & P_4 \end{bmatrix} = -\pi^2 \frac{P_t}{P_s} \frac{G(\pm 2iP_s)}{G(2 \pm 2iP_t)} \prod_{f \in F} \prod_{\substack{\sigma \in \mathbb{Z}_2^f \\ \sigma_f = -\eta_t(f)}} G\left(1 - i\sum_{i \in f} \sigma_i P_i\right)^{-\sigma_f} \frac{1}{\sqrt{\hat{d}}} \prod_{\substack{\sigma \in \mathbb{Z}_2^E \\ \sigma_V = 1}} \widetilde{G}\left(\hat{\omega}_{\epsilon} + \frac{i}{2}\sum_{i \in E} \sigma_i P_i\right)^{-\sigma_E}.$$

$$(4.3)$$

Compared to the original expression of [12], the most substantial change is our redefinition of $\hat{\omega}_{\pm}$, which we have shifted by $\frac{i}{2} \sum_{i \in E} P_i$. As a result, in the formula for the kernels $\hat{\mathbf{F}}^{\epsilon}_{P_s P_t}$ as well as in the definition (1.9) of $\hat{\omega}_{\pm}$, we have combinations of momentums of the type $\frac{i}{2} \sum_{i \in E} \sigma_i P_i$. Notice that $\hat{\omega}_{\pm}$ is only defined modulo integers: this does not affect the kernels, thanks to the identity $\widetilde{G}(x + 1) = -\frac{\pi}{\sin(\pi x)} \widetilde{G}(x)$.

**Fusion, analyticity and parity properties**

The existence of two kernels comes from the choice of a branch for the square root $\sqrt{\hat{d}}$. This square root appears explicitly in the formula for the kernels, and is also present in the definition (1.9) of $\hat{\omega}_\pm$, which is why we have two such quantities. Both kernels obey non-trivially the fusion relation (1.3) – as was checked numerically in [12] – where the integration contour has to be chosen such that it does not intersect the branch cuts, which is done by the shift $i\mathbb{R} \to i\mathbb{R} + \Lambda$ with $\Lambda$ large enough.

We can define a non-trivial single-valued, meromorphic quantity by adding the two branches. Since the factor $\sqrt{\hat{d}}$ changes sign, the single-valued combination is $\frac{1}{2}(\hat{\mathbf{F}}^+ - \hat{\mathbf{F}}^-)$. But this combination is zero when integrated against $t$-channel conformal blocks, because both kernels satisfy (1.3). In fact, we numerically find that this combination is an *odd* function of $P_s$ and $P_t$, which will turn out to coincide with the image $\mathfrak{R}\mathbf{F}$ of the $c = 25$ fusion kernel under Virasoro–Wick rotation.

On the other hand, we find numerically that the combination $\hat{\mathbf{F}} = \frac{1}{2}(\hat{\mathbf{F}}^+ + \hat{\mathbf{F}}^-)$ is an *even* function in $P_s$ and $P_t$, and we call it the *physical* fusion kernel at $c = 1$. Nevertheless, it obviously still has branch cuts.

For completeness, let us mention also how the kernels transform under the reflections $P_i \to -P_i$. The tetrahedral symmetry of (4.3) is all about permuting the momentums, but the behaviour under reflections is far from manifest. The physical fusion kernel $\frac{1}{2}(\hat{\mathbf{F}}^+ + \hat{\mathbf{F}}^-)$ must be invariant under reflections of all six momentums $P_1, P_2, P_3, P_4, P_s, P_t$, but this need not apply to the two kernels $\hat{\mathbf{F}}^+$ and $\hat{\mathbf{F}}^-$. Numerically, we find that $\hat{\mathbf{F}}^+$ and $\hat{\mathbf{F}}^-$ are invariant under reflections of $P_1, P_2, P_3, P_4$, but under reflections of $P_s$ or $P_t$ we have $\hat{\mathbf{F}}^\pm \to \hat{\mathbf{F}}^\mp$.

## 4.2 Two different formulas for the $c = 25$ fusion kernel

**Virasoro–Wick rotation of $c = 1$ kernels**

Let us now define two kernels $\mathbf{F}^\pm$ at $c = 25$ from the $c = 1$ kernels $\hat{\mathbf{F}}^\pm$ (4.3) via the relation (4.2). The Virasoro–Wick rotation involves the permutation $1 \leftrightarrow 3, s \leftrightarrow t$, which leaves the set of faces $F$ and the condition $\sigma_V = 1$ invariant, but changes $\eta_t(f) \leftrightarrow -\eta_t(f)$. We obtain

$$\mathbf{F}_{P_s P_t}^{\epsilon=\pm} \begin{bmatrix} P_2 & P_3 \\ P_1 & P_4 \end{bmatrix} = \frac{\pi^2}{i} \frac{G(\pm 2P_t)}{G(2 \pm 2P_s)} \prod_{f \in F} \prod_{\substack{\sigma \in \mathbb{Z}_2^f \\ \sigma_f = \eta_t(f)}} G\left(1 + \textstyle\sum_{i \in f} \sigma_i P_i\right)^{-\sigma_f} \frac{\epsilon}{\sqrt{d}} \prod_{\substack{\sigma \in \mathbb{Z}_2^E \\ \sigma_V = 1}} \widetilde{G}\left(\omega_\epsilon - \tfrac{1}{2}\textstyle\sum_{i \in E} \sigma_i P_i\right)^{-\sigma_E} . \quad (4.4)$$

We then define $\mathbf{F} = \frac{1}{2}\sum_\pm \mathbf{F}^\pm$, which is given by our formula (1.7). The kernel $\mathbf{F}$ is now a meromorphic function of momentums, because we obtain it by summing over the two possible determinations of $\sqrt{d}$. Moreover, we numerically confirm that it is an *even* function of the momentums $P_s, P_t$ since under the corresponding reflections we observe that $\mathbf{F}^\pm \to \mathbf{F}^\mp$.

To finally check that $\mathbf{F}$ is indeed the physical fusion kernel at $c = 25$, we have numerically tested that it satisfies the fusion relation (1.3).

**Comparison with the Teschner–Vartanov formula**

Let us specialize the Teschner–Vartanov formula for the fusion kernel (2.17) to the case $c = 25$ i.e. $b = 1$. The relevant special functions reduce to combinations of Barnes' $G$-function,

$$\Gamma_1(x) = \frac{(2\pi)^{\frac{x-1}{2}}}{G(x)}, \qquad S_1(x) = (2\pi)^{x-1}\widetilde{G}(1-x). \quad (4.5)$$

This leads to

$$\mathbf{F}^{(b\to 1)}_{P_s P_t}\left[\begin{smallmatrix} P_2 & P_3 \\ P_1 & P_4 \end{smallmatrix}\right] = \frac{1}{8\pi^2}\frac{G(\pm 2P_t)}{G(2\pm 2P_s)}\prod_{f\in F}\prod_{\substack{\sigma\in\mathbb{Z}_2^f \\ \sigma_f=\eta_t(f)}} G\left(1+\sum_{i\in f}\sigma_i P_i\right)^{-\sigma_f}\int_{i\mathbb{R}}\frac{du}{i}\prod_{\substack{\sigma\in\mathbb{Z}_2^E \\ \sigma_V=1}}\widetilde{G}\left(u+\frac{\sigma_E}{2}+\frac{1}{2}\sum_{i\in E}\sigma_i P_i\right)^{\sigma_E}.$$

(4.6)

Comparing with (1.7), we observe that the main prefactors in the two expressions are the same, as required by tetrahedral symmetry. Then, the equality of the two formulas boils down to the identity

$$\int_{i\mathbb{R}} du\prod_{\substack{\sigma\in\mathbb{Z}_2^E \\ \sigma_V=1}}\widetilde{G}\left(u+\frac{\sigma_E}{2}+\frac{1}{2}\sum_{i\in E}\sigma_i P_i\right)^{\sigma_E} = 4\pi^4\sum_{\epsilon=\pm}\frac{\epsilon}{\sqrt{d}}\prod_{\substack{\sigma\in\mathbb{Z}_2^E \\ \sigma_V=1}}\widetilde{G}\left(\omega_\epsilon-\frac{1}{2}\sum_{i\in E}\sigma_i P_i\right)^{-\sigma_E}.$$

(4.7)

(In the special case $P_1 = P_2 = P_3 = P_4 = \frac{1}{4}$, a similar simplification of the Ponsot–Teschner formula for the fusion kernel was already observed in [12].) We have checked this identity numerically and there is little doubt that it is true. While an analytic proof is currently lacking, it could definitely lead to valuable insights, especially if it led to a generalization beyond the case $c = 25$. The theory of elliptic hypergeometric integrals [35] may be relevant, since the Teschner–Vartanov integral is a limit of such integrals.

## Acknowledgements

We are grateful to Dionysios Anninos, Panos Betzios, Scott Collier, Lorenz Eberhardt, Cristoforo Iossa, Oleg Lisovyy, Alexander Maloney, Eric Perlmutter, and Julien Roussillon, for fruiful discussions and correspondence. We thank Lorenz Eberhardt, Oleg Lisovyy and Eric Perlmutter for helpful comments on the draft text.

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
