# Peer review of "On the Virasoro fusion kernel at $c=25$"

_SciPost Physics, doi:SciPost Phys. 17, 058 (2024)_

## Round 2 · Referee Report · Anonymous (Referee 1) · 2024-7-22

Strengths

  1. Well-written.
  2. Propose a new formula for c=25 Virasoro fusion kernel. Very interesting results.
  3. Clear derivation and proof.

Weaknesses

/

Report

See attached report.

Attachment

Recommendation

Publish (easily meets expectations and criteria for this Journal; among top 50%)

  • validity: top
  • significance: top
  • originality: top
  • clarity: top
  • formatting: perfect
  • grammar: perfect

Author:  Ioannis Tsiares  on 2024-07-29  [id 4662]

(in reply to Report 1 on 2024-07-22)
Category:
answer to question

Dear referee,

Thank you for your time and your useful comments.

We address the three points you raised at the end of your report below:

  1. In our (2.17), the integral factor indeed has tetrahedral symmetry, as it is invariant under any permutation of the edges that preserves the vertices. On the other hand, the prefactor singles out the $t$-edge. With a different prefactor, we could obtain 6j symbols that respect tetrahedral symmetry, as in Eq. (2.27) of ref. [11]. Our prefactor follows from the natural normalization (1.10) of conformal blocks.

  2. The scenario is that for generic $c$ the infinite sum is obtained from the integral expression by rewriting it as sum of residues of its poles. It is not straightforward to do this, as we cannot move the integration contour to infinity, but this may be possible with the help of the shift equations for the integrand. Then, for $c\to (-\infty, 1)$, the integral expression no longer makes sense, and the infinite sum becomes divergent, and may be interpreted as a distribution. For $b^2\in \mathbb{Q}$, we indeed expect a similar picture as for $b=i$ and $b=1$. Again, this would require nontrivial analytic manipulations of the integral formulas.

  3. Assuming that the physical kernels for $c\leq 1$ are distributions, there are, in principle, no new surprising implications for the inner product of conformal blocks: the inner product, just like the kernels, should always be thought of as integrated against a test function (i.e. a conformal block) to yield a physical observable and hence it is OK if it happens to be a distribution as a function of $P$ (the integrated momentum). On the other hand, in the recent work by Collier, Eberhardt, and Zhang it was shown that for $c>25$, the space of Liouville conformal blocks forms a Hilbert space of a 3d TQFT that corresponds to the quantization the Teichmuller space of Riemann surfaces. The corresponding inner product takes the form of an integral over Teichmuller space with a measure given by the bc ghosts and a time-like Liouville partition function. It is unclear to us whether there exists a well-defined TQFT along the lines of this work for $c\leq 1$, and whether the corresponding inner product would take an analogous form (i.e. as an integral over Teichmuller space) assuming that it's a distribution. Deriving rigorously the $c\leq 1$ kernels from the CFT perspective would definitely shed light on these interesting questions.

Kind regards, The authors

---

## Round 2 · Referee Report · Anonymous (Referee 2) · 2024-7-31

Strengths

1- A new formula for the fusion kernel for c=25 Virasoro conformal blocks suggested using the introduced Virosoro-Wick transformation, 2- Numerical check of coincidence of new fusion kernel for c=25 with the Teschner-Ponsot fusion kernel.

Weaknesses

1- There is no analytic proof for the coincidence of the proposed fusion kernel for c=25 with the Teschner-Ponsot fusion kernel. It is not critical for publishing the paper, but it seems to be a good task for the future since it may reveal some important mathematical structures.

Report

The paper "On the Virasoro fusion kernel at c=25" by Sylvain Ribault and Ioannis Tsiares is devoted to studying the fusion kernels of Virasoro conformal blocks. The fusion kernels are kernels of integral transformations relating 4-point Virasoro conformal blocks in s- and t- channels. These two channels correspond to two bases in the space of conformal blocks associated with different asymptotic regions in the space of positions of four fields. The fusion kernels relating these two bases can be considered as a deep generalization of connection matrices for hypergeometric series.

Teschner and Ponsot found an explicit formula for the fusion kernel valid for any real central charge $c>1$ (and for central charges with non-zero imaginary parts). Later, a formula for the fusion kernel for $c=1$ was proposed by Iorgov, Lisovyy, and Tykhyy relating the fusion kernel with the connection constant of Painleve VI equation.
These types of formulas for the fusion kernel are given for different values of cental charge and look very different.
Sylvain Ribault and Ioannis Tsiares in this paper propose a so-called Virosoro-Wick transformation for the fusion kernels relating central charges $c$ and $26-c$.
It allowed them to suggest an alternative formula for the fusion kernel for $c=25$ using the fusion kernel for $c=1$ and to check numerically its coincidence with Teschner-Ponsot formula.
For the future, it is interesting to find an analytic proof of this coincidence and to try to find a generalization of this story to some other central charges.

I recommend the paper "On the Virasoro fusion kernel at c=25" by Sylvain Ribault and Ioannis Tsiares for publication in SciPost.

Recommendation

Publish (easily meets expectations and criteria for this Journal; among top 50%)

  • validity: high
  • significance: high
  • originality: high
  • clarity: high
  • formatting: excellent
  • grammar: good

Author:  Ioannis Tsiares  on 2024-08-02  [id 4675]

(in reply to Report 2 on 2024-07-31)

Dear referee,

Thank you for your time and the comments in your report.

Regarding the weakness comment, we'd like to mention that we have an upcoming publication that explains the analytic origin of the formula for the c=25 fusion kernel.

Kind regards,
The authors.

---

## Editorial Decision

published